# The First Record of Taiga Shrew in Lithuania

**DOI:** 10.3390/ani15213088

**Published:** 2025-10-24

**Authors:** Linas Balčiauskas

**Affiliations:** State Scientific Research Institute Nature Research Centre, Akademijos 2, 08412 Vilnius, Lithuania; linas.balciauskas@gamtc.lt

**Keywords:** *Sorex isodon*, mammal fauna, distribution range

## Abstract

**Simple Summary:**

We report the first record of the taiga shrew (*Sorex isodon*) in Lithuania, expanding its known distribution westward into the Baltic region. The specimen was collected in 2023 during non-systematic trapping and identified through external morphology and cranial traits. Identification relied on tail structure and distinctive skull features, with confirmation supported by expert consultation. This finding suggests possible habitat suitability and underreported presence of the species in Lithuania. It also highlights the need for further surveys, re-evaluation of skull collections, and genetic identification of the species to better understand its presence in the region.

**Abstract:**

The taiga shrew (*Sorex isodon*), a boreal forest species widely distributed across northern Eurasia, had not previously been recorded in the Baltic states. We report the first occurrence of *S. isodon* in Lithuania, which extends the species range westward into the Baltic region. The specimen, which was initially misidentified as *Sorex araneus*, was collected in 2023 near Vilnius during non-systematic snap-trapping. A detailed examination of tail, cranial, and dental characteristics confirmed the identification as *S. isodon*. The individual was a non-adult male, weighing 5.8 g, and was trapped in a wet mixed forest influenced by beaver activity. This suggests that the microhabitat conditions are similar to those reported in Belarus and Ukraine. This finding extends the western boundary of *S. isodon* distribution by approximately 200–630 km. This study underscores the potential underreporting of this species in the Baltic region and highlights the importance of revisiting existing skull collections, conducting targeted trapping, and performing genetic analyses. Verifying additional records will clarify the habitat preferences and conservation relevance of *S. isodon* at the western edge of its range.

## 1. Introduction

The taiga shrew (*Sorex isodon* Turov 1924), also called the even-toothed shrew [1], can be found in Russia, China, Mongolia, and Kazakhstan, as well as in the eastern part of Europe [2]. The distribution range of this species extends from Fennoscandia and Central Russia [3] to the northern part of Belarus and the northern part of Ukraine [4,5,6].

The first record of *S. isodon* in Belarus was from 1996 in the Brest Region, Berezinsky Biosphere Reserve [4]. Later some specimens were repeatedly recorded from the same area [4], and to the north of it [5]. In the north of Belarus, a single individual was captured in 2020 in the Vitebsk Region [6], this being the northernmost record of the species in the country. In the north of Ukraine, S. *isodon* was first found in the Desnyansko-Starogutsky National Nature Park, 2000–2004 [5]. All of these records (Figure 1) were connected to the swampy habitats related to river valleys [4,5,6]. Further investigation and revision of museum collections were recommended by A. Savarin [6] to further clarify species distribution in the region.

In Fennoscandia, *S. isodon* was first recorded in 1949 [3,7]. In Finland, over 70 localities of the species are known, but in Sweden and Norway species distribution is patchy, with two isolated populations in both countries [8,9,10]. One isolated record of *S. isodon* is known from the northern part of Norway, but later findings have not confirmed this record [11]. In short, the taiga shrew is a widespread but locally rare habitat specialist with small, isolated, and vulnerable populations outside Russia.

Distribution of *S. isodon* in the European and Asian parts of Russia is wide and patchy [12,13,14,15,16]. Shrews avoid human-transformed habitats, preferring mature coniferous forests rich in herbs [15]; however, in the northern and northwestern parts of their distribution range adaptations may have occurred [14]. In Central Russia, *S. isodon* is the least abundant of sympatric Soricidae species [13], the same in Karelia, NW Russia, where the species is largely stenotopic [14]. In the NE of the European part of Russia, these shrews are abundant in mountainous spruce forests, even dominating the community [13].

Habitat specialization is characteristic of *S. isodon* not only in the boreal forests of the European part of Russia [13,14], but also in Siberia. In Central Siberia, the species is associated with dark spruce–fir forests, dense ground vegetation, and high humidity [17]. In the eastern part of Siberia, the Upper Angara Basin, larch and mixed forests with rich undergrowth are favored, while dry pine or open habitats are avoided [18]. The mosaics of rich spruce forests, lush meadows, wetlands, and riparian zones are favored by *S. isodon* in Finland [19,20]. Norwegian habitats additionally include overgrown meadows and rocky slopes [10]. All species findings in Ukraine and Belarus are related to swampy areas [4,5,6]; therefore, *S. isodon* has strict habitat requirements, primarily favoring cool, moist, mature coniferous forests, and is largely restricted to undisturbed or minimally disturbed forest ecosystems.

Despite being part of multispecies shrew communities, including up to six syntopic [21] and eight sympatric [22]. Sorex species, *S. isodon* maintains ecological separation. Their interactions with other shrew species vary significantly across regions and habitats.

While *Sorex araneus*, *S. caecutiens*, *S. minutus*, and *S. isodon* can coexist in the same regions, they differ in habitat preferences, demography, and social behavior, having the lowest proportion, fewer interactions and behavior that is more aggressive [13]. Trophic studies suggest *S. isodon* selects prey by size, favoring larger soil-dwelling invertebrates compared to smaller, surface-dwelling prey preferred by species like *S. minutus* [23]. The guild of forest shrews with syntopic *S. isodon* in Siberia may include *S. caecutiens*, *S. roboratus*, and *S. minutissimus*, as well as more generalist or meadow-associated species like *S. minutus* and *S. tundrensis* [24]. In Central Russia, *S. isodon* overlaps with *S. araneus*, *S. minutus*, and *S. caecutiens* [25]. In the Far East and Mongolia, composition of the shrew community differs, including species not present in the western part of *S. isodon* distribution [26,27]. The share of *S. isodon* in multi-species shrew communities is usually less than 10% [21,22].

Prey preferences of *S. isodon* overlap with other shrews, as size-based selectivity contributes to dietary niche partitioning [23]. Being of medium size, *S. isodon* can consume larger prey than small shrews (e.g., *S. minutus*. *S. minutissimus*, and *S. caecutiens*) [28]. Most frequent prey of *S. isodon* are larvae of various insects, spiders, beetles, and earthworms, this last item characterized by the highest intake compared to other shrews [28]. Compared to other shrews, *S. isodon* is also characterized by more frequent soil foraging, compared to the surface-foraging of other species [22,23,28,29].

Globally characterized as a Least Concern species [20], *S. isodon* was classified as a Data Deficient species in Norway [30]. This situation remains, as new records are related to the western part of the country only [10]. The species is not threatened in other countries [30].

The aim of our study is to report first record of *S. isodon* in Lithuania, as this finding not only expands the list of small mammal species in the Baltic region [31], but also extends the species distribution range in Eastern Europe westward.

## 2. Materials and Methods

### 2.1. Study Site

The study site is located in the outskirts of Vilnius, the capital city of Lithuania (Figure 2). Since small mammals were trapped not only in the forest near a residential area but also near residential houses and in one of their yards, precise details are not provided due to personal data protection.

Small mammals were trapped in commensal habitats (the lawn of the house, outbuildings, and garage), in the hedge next to the driveway of the residential buildings, and in the forest. The forest habitat is a mixed forest with spruce trees and is wet with a high water level (maintained by beaver activity).

### 2.2. Small Mammal Trapping

During July to September 2023, a non-professional individual collected a sample of small mammals using an unsystematic snap-trapping method. The trapping effort (120 trap days) was not fully documented, and the sample was composed of animals from several habitats pooled together. The entire sample (53 individuals) was stored under refrigeration. Our attention to this sample was granted by an unexpectedly high number of water shrews (*Neomys fodiens*); therefore, all small mammals were analyzed (i.e., species identification, body mass and body length to calculate body condition index, and dissection to obtain samples for pathogen research).

### 2.3. Identification of Species

The specimen (field label GTC-2023-643) was a medium-sized *Sorex* which attracted attention due to unusual coat color, tail shape, and long claws. The individual was originally misidentified as *S. araneus*. Despite this, the skull was preserved, cleaned using *Dermestes* beetles, and later examined under binoculars. Unfortunately, no pictures of the individual were taken to assess pelt color and body proportions, and the individual was not preserved, as we did not expect *S. isodon* in Lithuania. Therefore, genetic identification was not available. The skull was damaged in the braincase area by a snap trap; however, the rostrum part and the lower jaw were suitable for analysis.

To identify our specimen and to confirm this is not *S. araneus*, we used several references, indicating differences in external, cranial, and dental traits [6,9,10,32,33].

The coat color of *S. isodon* is reported as brown with a pinkish tinge [6]. The tail is square-shaped or four-angled—this feature easily grasped with the fingers—and has a tuft of longer hair at the end [6,9]. Body length ca. 61 mm [6], but variation is high, from 40 to 87 mm [9]. Condylobasal length of the skull is between 17.7 and 20.2 mm and the width of the skull between 9.0 and 10.3 mm [9].

Dental characters of *S. isodon* also have some differences from *S. araneus.* In *S. isodon*, the fifth intermediate tooth in the upper jaw is well developed and intensely pigmented [6,9,10,32]. Unicuspid size gradually decreases from front to back.

In *S. isodon*, the *foramen mentale* relative to the first molar and premolar is positioned closer to the front of the mandible than in *S. araneus*, in which the *foramen mentale* is located further back [6,9,32]. The temporal fossa has a distinct triangular shape [33], with an outer angle > 100° and an inner angle < 60°. Finally, the coronoid process in *S. isodon* is more rounded than that in *S. araneus* [6,33].

### 2.4. Sample Processing

Before dissection, all small mammals of the investigated sample were weighed to the nearest 0.1 g using electronic scales. The body length (L) was measured in standard Morrison–Scott method, from the end of snout to anus opening [34], to the nearest 0.1 mm using a mechanical caliper. The body condition index (BCI) was calculated as in Moors [35], to maintain compatibility with the previous research [36]. Gender of shrews was determined by distance between the urogenital and excretory openings, wherein a “short distance between the two openings indicating a female and the openings occurring within the same hair-free patch” [37]. The details of age determination are presented in [36]; however, for *S. isodon*, it was limited due to the damage caused by a snap-trap.

To characterize the diversity of small mammals in the analyzed sample, I used a diversity index, Shannon’s H, using log_2_ base [38]. Calculation was performed with PAST, version 5.2/2 (Paleontological Museum, University of Oslo, Oslo, Norway) [39].

The distances to the nearest records of *S. isodon* in Belarus and Ukraine were calculated from coordinates, using the Haversine formula [40].

Photographs of the skull (field label GTC-2023-643, now identified as *S. isodon*), and the skull of *S. araneus* for comparison of traits, were taken with a Canon EOS 6D camera and Canon MP-E65 mm macro lens (Canon, Tokyo, Japan) using a MJKZZ automated focus stacking rail set (MJKZZ.de, Vienna, Austria). The obtained image layers were stacked using the program ZereneStacker 1.04 (https://zerenestacker.com/cms/about_us, accessed on 17 September 2025).

## 3. Results

The analyzed specimen was a non-adult male, with a body mass of 5.8 g, and body length of 61.5 mm. The tail was angular to the touch, with a tuft of longer hair at the end. The long claws on the front and hind legs also attracted attention. The age of the animal could not be accurately determined because the skull and neck, as well as the thyroid gland, had been damaged by a trap.

After cleaning, dentition and skull characteristics confirmed species identification as *Sorex isodon*, previously unrecorded in Lithuania. The fifth unicuspid tooth in the upper jaw is well developed and has a dark top (Figure 3a). All unicuspid teeth are visible, with no clear size differences (Figure 3b). On the mandible, position of *foramen mentale* is characteristic of *S. isodon*, not *S. araneus* (Figure 3c). The temporal fossa is triangular in shape (Figure 3d). Inner shape of *processus condylicus* is less concave (Figure 3e) than in *S. araneus*. These traits confirm species identification. Comparison to *S. araneus* skull is presented as Appendix A.

The investigated sample consisted of 53 individuals from 7 species: 21 striped field mice (*Apodemus agrarius*), 12 house mice (*Mus musculus*), 11 *S. araneus*, 6 *N. fodiens*, 1 common vole (*Microtus arvalis* sensu lato), 1 birch mouse (*Sicista betulina*) and 1 *S. isodon*. All specimens were collected from the same location as the *S. isodon* specimen during the same trapping effort. Considering the limited trapping effort, the sample was characterized by high diversity (Shannon’s H = 2.247) and low dominancy.

As for body mass and body length of *S. isodon* compared to *S. araneus*, it was smaller: Q = 7.2–12.2 vs. 5.8 g, and L = 56.1–73.3 vs. 61.5 mm. The body condition index was mostly lower: BCI = 2.39–4.47 in *S. araneus* vs. 2.49 in *S. isodon*.

## 4. Discussion

During the last few decades, new small mammal species have been discovered in Lithuania. The pygmy field mouse (*Apodemus uralensis*) was first trapped in 1997 but identified in 1999 using morphological characters [41]. The sibling vole (*Microtus rossiaemeridionalis*) was identified in 1999 using karyological methods, but later identified in the skull collections of previous years [42]. Similarly, the Miller’s shrew (*Neomys milleri*), was first identified in 2009, and then later identified from skulls of individuals trapped in 2001 and 2008 [43]. Afterwards, the presence of *N. milleri* was confirmed in Estonia from skulls collected in 1981–1989 [44].

Similarly in Latvia, the European pine vole (*Microtus subterraneus*), identified genetically in 2006 [45], and the root vole (*Microtus oeconomus*), trapped in 1989 but described in 2014 [46], both were later confirmed quite abundantly in the materials of owl pellets [31].

The description of the mammals in the countries and regions of Europe had been occurring for well over a century [47]; however, they were only considered subspecies according to the understanding of the time. New records of species and expanding distribution ranges have also been reported in the last few decades. For example, the greater white-toothed shrew (*Crocidura russula*) was recorded in Ireland [48], Great Britain [49], Czech Republic [50], and Fennoscandia [51]. Records, expanding the geographical range of various species, were also reported for rodents, such as the common vole (*Microtus arvalis*) and European snow vole (*Chionomys nivalis*) in Portugal [52,53], Bavarian pine vole (*Microtus bavaricus*) in Croatia [54], and *M. subterraneus* in Mordovia, Russia [55]. Previously unrecorded species for specific countries have also been discovered from owl pellets, as was the case with the mouse-tailed dormouse (*Myomimus roachi*) in Greece [56].

The habitat of *S. isodon* in Lithuania is not known, although the single specimen was trapped in a wet mixed forest. Given water levels were regulated by beavers (*Castor fiber*), the site was not clearly visible on the presented map (see Figure 2). Swampy habitats were also reported in records from Belarus and Ukraine [4,5,6].

On the other hand, shrews can unexpectedly be found in quite unusual habitats, that can be characterized as commensal. For instance, the bicolored shrew (*Crocidura leucodon*) was accidentally snap-trapped in a barn, on the outskirts of Hamburg, Germany [57]. Former investigation has shown that records of *N. fodiens* and *N. milleri* in commensal habitats of Lithuania, such as outbuildings in homesteads, represent a new ecological feature observed in these species [58].

The first record of *S. isodon* in Lithuania is in 200–225 km from the nearest localities known in Belarus and 630 km from a locality in Ukraine [4,5,6]. The Lithuanian point is west of all three of the others, by 0.52–3.46° from those in Belarus and 8.55° in Ukraine. 

As for further steps, additional trapping in the site is required to confirm the presence of *S. isodon*, as is taking pictures of the animal to assess pelt color and body proportions. The genetic identification of newly trapped specimens is recommended, as well as the review of the shrew skull collection at the State Scientific Research Institute Nature Research Centre, Lithuania. The skull collection of small mammals is currently maintained by the Mammal Ecology Laboratory. In the future, the Nature Research Centre plans to complete the installation of the Collections Department for the storage of all available collections.

## 5. Conclusions

The discovery of *Sorex isodon* in Lithuania represents a westward expansion of the species’ known range in central-eastern Europe.

## Figures and Tables

**Figure 1 animals-15-03088-f001:**
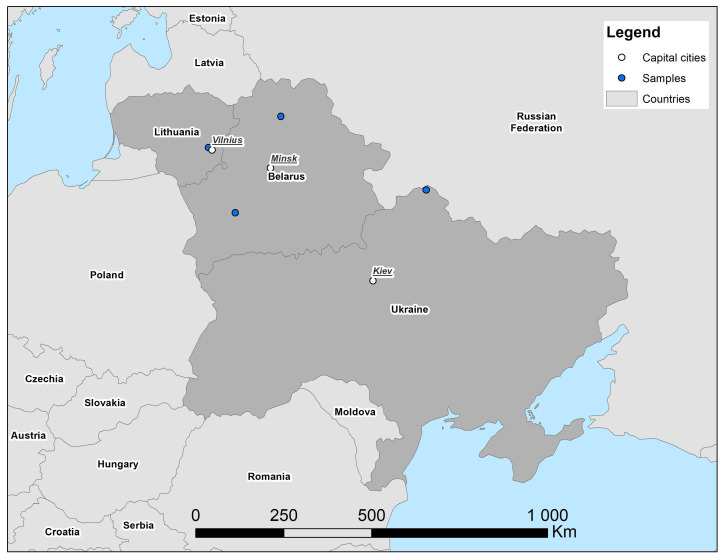
Records of *Sorex isodon* in Belarus and Ukraine, 1998–2020, after [4,5,6], and new record for Lithuania in 2023. No further records available after 2020 from this region. Delineation of state borders is not intended to imply political stance of author or publisher.

**Figure 2 animals-15-03088-f002:**
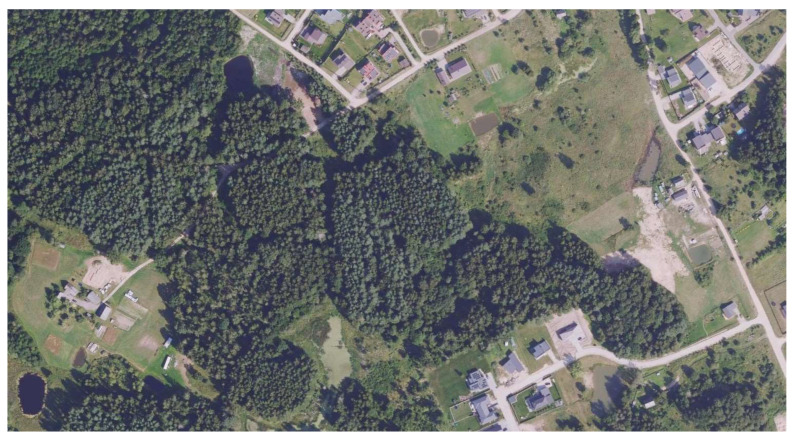
Study Site (54.759431° N, 25.143096° E). Source: Public Institution Construction Sector Development Agency. (2025). Geographic data services. Lithuanian Spatial Information Portal (Geoportal), https://www.geoportal.lt/arcgis/rest/services/NZT (accessed on 25 September 2025).

**Figure 3 animals-15-03088-f003:**
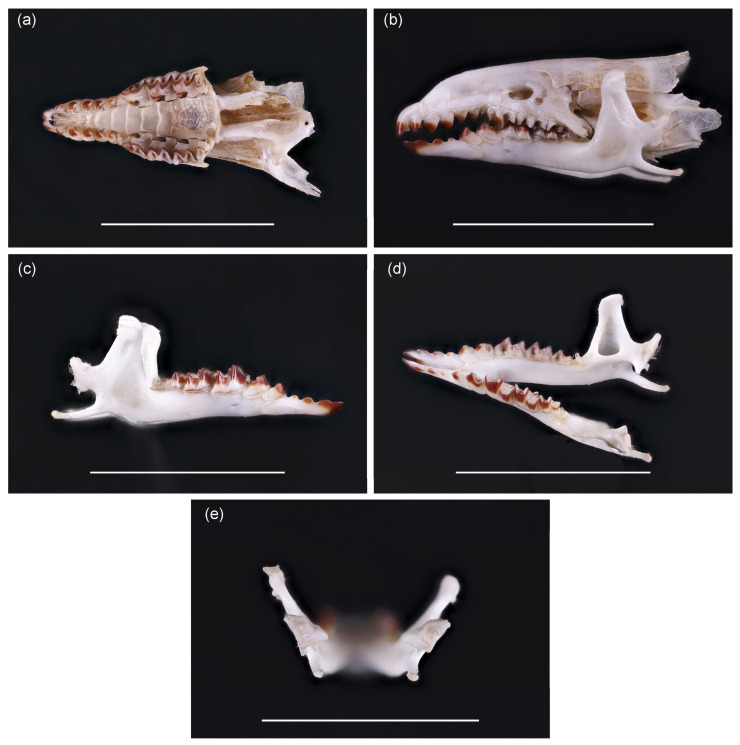
Skull of *S. isodon* trapped in Lithuania: (**a**) upper jaw, from the ventral side of the skull; (**b**) skull view from the side, indicating the height of unicuspids, (**c**) position of the *foramen mentale* on the mandible, (**d**) shape of the temporal fossa, and (**e**) *processus condylicus*. Scale bar is equal to 1 cm. For more precise references to the location of these features in the skull and comparison with the closest species, *Sorex araneus*, refer to Appendix A.

## Data Availability

No data other than presented are available.

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
