# Peer review of "The First Record of Taiga Shrew in Lithuania"

_animals, 2025, doi:10.3390/ani15213088_

Round 1
Reviewer 1 Report
Comments and Suggestions for Authors
Dear authors, The submitted manuscript is sensational, as it expands our understanding of the taiga shrew's range. Its discovery in Lithuania is unexpected. This should be considered a first for the country, as no previous sightings of the species have been reported in the literature available to me. The manuscript includes images of the skull, which are sufficient to confirm the animal's species identity. Furthermore, a detailed analysis of possible commensal relationships with other shrew species across most of its range is provided. Therefore, further study of the species in Lithuania is truly important to determine its status among small insectivores. It will be particularly interesting to know how frequently this shrew is found compared to the Eurasian common and Eurasian pygmy shrews. I recommend this publication for journal Animals.
Author Response
Rev#1 comments and answers
( ) I would not like to sign my review report
(x) I would like to sign my review report
Quality of English Language
( ) The English could be improved to more clearly express the research.
(x) The English is fine and does not require any improvement.
|
Yes |
Can be improved |
Must be improved |
Not applicable |
|
|
Does the introduction provide sufficient background and include all relevant references? |
(x) |
( ) |
( ) |
( ) |
|
Is the research design appropriate? |
(x) |
( ) |
( ) |
( ) |
|
Are the methods adequately described? |
(x) |
( ) |
( ) |
( ) |
|
Are the results clearly presented? |
(x) |
( ) |
( ) |
( ) |
|
Are the conclusions supported by the results? |
(x) |
( ) |
( ) |
( ) |
|
Are all figures and tables clear and well-presented? |
(x) |
( ) |
( ) |
( ) |
Comments and Suggestions for Authors
Cpmment: Dear authors, The submitted manuscript is sensational, as it expands our understanding of the taiga shrew's range. Its discovery in Lithuania is unexpected. This should be considered a first for the country, as no previous sightings of the species have been reported in the literature available to me. The manuscript includes images of the skull, which are sufficient to confirm the animal's species identity. Furthermore, a detailed analysis of possible commensal relationships with other shrew species across most of its range is provided. Therefore, further study of the species in Lithuania is truly important to determine its status among small insectivores. It will be particularly interesting to know how frequently this shrew is found compared to the Eurasian common and Eurasian pygmy shrews. I recommend this publication for journal Animals.
Answer: thank you
We cannot indicate, how frequently this shrew is in comparison to other shrew species in Lithuania. In multi-species shrew community of Siberia, S. isodon is not dominant, usually comprising < 10 %.
Reviewer 2 Report
Comments and Suggestions for Authors
The article "The first record of taiga shrew in Lithuania" is a short but very solid scientific communication.
This publication presents the first confirmed record of the taiga shrew (Sorex isodon) in Lithuania, extending the species' known range by approximately 200–630 km westward. The work is of significant faunal importance and complements data from Belarus and Ukraine. It is logical, coherent, and well-documented.
Documentation of diagnostic features (e.g. shape of foramen mentale, triangular temporal fossa, tooth pigmentation) is consistent with the literaturę. Znaczenie faunistyczne i porównawcze zostało dobrze wyważone – praca nie wyciąga nadmiernych wniosków z pojedynczego osobnika.
My comments: 1. Lack of photographic documentation of the specimen in vivo.
The author admits that no photographs were taken in the field—this is worth mentioning in the Discussion as a limitation affecting the ability to assess color characteristics and body proportions. 2. Morphological identification only.
The lack of genetic confirmation was noted. In the future, the author should emphasize that a review of the collection and DNA testing are key to fully confirming the species' presence (which he himself recommends).
3. Terminological inconsistency: The title and abstract use the phrase "first record of taiga shrew," while the text sometimes uses "new species for the country." The latter form could be interpreted as a scientific discovery—it's worth consistently using "first record for Lithuania." 4. Lack of information about the skull's deposit. The location of the specimen (museum, catalog number) should be added—currently, only the "field label GTC-2023-643" is used. 5. Minor language errors:
o "as this finding not only adds new species" → as this finding not only adds a new species o "commensal habitats ... are a new feature of their ecology" → better: represent a new ecological feature observed in these species.
Notes regarding Figure 2: • Base boundary layers on a standard base (e.g., Natural Earth 1:10m political boundaries),
• Avoid drawing lines dividing disputed territories,
• Add a simple, non-political note to the map caption, e.g., "The state borders shown on the map are approximate and do not reflect the authors' political stance."
Maybe it should be added? -The delineation of boundaries does not imply any political position of the author or the publisher.
In summary: • This work has clear documentary value. It contributes an important element to the discussion of the range of Sorex isodon in Eastern Europe, confirming the trend of western expansion and potential range shifts in the context of climate change. • Recommended for acceptance after minor linguistic corrections and supplementation of information on the specimen's deposit.
Comments on the Quality of English LanguageRecommended for acceptance after minor linguistic corrections
Author Response
Rev#2 comments and answers
( ) I would not like to sign my review report
(x) I would like to sign my review report
Quality of English Language
(x) The English could be improved to more clearly express the research.
( ) The English is fine and does not require any improvement.
|
Yes |
Can be improved |
Must be improved |
Not applicable |
|
|
Does the introduction provide sufficient background and include all relevant references? |
( ) |
(x) |
( ) |
( ) |
|
Is the research design appropriate? |
( ) |
(x) |
( ) |
( ) |
|
Are the methods adequately described? |
( ) |
(x) |
( ) |
( ) |
|
Are the results clearly presented? |
( ) |
(x) |
( ) |
( ) |
|
Are the conclusions supported by the results? |
( ) |
(x) |
( ) |
( ) |
|
Are all figures and tables clear and well-presented? |
( ) |
(x) |
( ) |
( ) |
Comments and Suggestions for Authors
The article "The first record of taiga shrew in Lithuania" is a short but very solid scientific communication.
This publication presents the first confirmed record of the taiga shrew (Sorex isodon) in Lithuania, extending the species' known range by approximately 200–630 km westward. The work is of significant faunal importance and complements data from Belarus and Ukraine. It is logical, coherent, and well-documented.
Documentation of diagnostic features (e.g. shape of foramen mentale, triangular temporal fossa, tooth pigmentation) is consistent with the literaturę. Znaczenie faunistyczne i porównawcze zostało dobrze wyważone – praca nie wyciąga nadmiernych wniosków z pojedynczego osobnika.
Comment 1. Lack of photographic documentation of the specimen in vivo.
The author admits that no photographs were taken in the field—this is worth mentioning in the Discussion as a limitation affecting the ability to assess color characteristics and body proportions.
Answer: we added text “, taking pictures of the animal to assess pelt color and body proportions”
Comment 2. Morphological identification only.
The lack of genetic confirmation was noted. In the future, the author should emphasize that a review of the collection and DNA testing are key to fully confirming the species' presence (which he himself recommends).
Answer: done, see Lines 231-232.
Comment 3. Terminological inconsistency: The title and abstract use the phrase "first record of taiga shrew," while the text sometimes uses "new species for the country." The latter form could be interpreted as a scientific discovery—it's worth consistently using "first record for Lithuania."
Answer: Thank you, we changed text to avoid misunderstanding.
Line 97 “not only expands the list of small mammal species in the Baltic region”
Line 174 “previously not recorded in Lithuania”
Line 214 “Previously not recorded species for the country”
- Lack of information about the skull's deposit. The location of the specimen (museum, catalog number) should be added—currently, only the "field label GTC-2023-643" is used.
Answer: small mammal skull collection is not in a museum state at a moment. We add text “The skull collection of small mammals is currently maintained by the Mammal Ecology Laboratory. In the future, the Nature Research Centre plans to complete the installation of the Collections Department for the storage of all available collections.”
- Minor language errors:
"as this finding not only adds new species" → as this finding not only adds a new species
"commensal habitats ... are a new feature of their ecology" → better: represent a new ecological feature observed in these species.
Answer: line 96, after the edit first comment is no longer actual, as the text is “as this finding not only expands the list of small mammal species in the Baltic region”
Second comment – text changed as recommended.
Notes regarding Figure 2: • Base boundary layers on a standard base (e.g., Natural Earth 1:10m political boundaries),
- Avoid drawing lines dividing disputed territories,
- Add a simple, non-political note to the map caption, e.g., "The state borders shown on the map are approximate and do not reflect the authors' political stance."
Maybe it should be added? -The delineation of boundaries does not imply any political position of the author or the publisher.
Answer: Reviewer meant Figure 1, not 2. We added required text as “The delineation of state borders is not intended to imply the political stance of the author or the publisher.”
In summary: • This work has clear documentary value. It contributes an important element to the discussion of the range of Sorex isodon in Eastern Europe, confirming the trend of western expansion and potential range shifts in the context of climate change. • Recommended for acceptance after minor linguistic corrections and supplementation of information on the specimen's deposit.
Answer: thank you for your comments, we accepted all of them.
Reviewer 3 Report
Comments and Suggestions for Authors
The author does a good job at their intended goal. This represents a new record expanding the known range of a species, and providing at least some context for it.
Having said that, there are a few small ways the paper could be improved, and a few things that need correcting. I have included an attached annotated PDF pointing these things out more directly in the text.
Overall, there are a few places that need fixed grammatically. These mostly deal with the use of articles, and, more often than not, are from missing articles. I have pointed some of these out in the PDF.
I agree with the identification, so the main thing is providing some clarification for the reader in the paper.
In particular, there is an instance that needs clarification in the Materials and Methods section on what is meant by 'no pictures taken.'
Additionally, and perhaps what needs the most work, are a few things in the Results section. These are in the annotated PDF, but there are several things that either need to be further explained or clarified. This includes Figure 3, which, while I know some features are pointed out in Figure S1, still needs to have some edits done. Some of these features should be pointed out, or labeled, or both, and this is made more clear that the caption states the parts of the figure are supposed to highlight particular features, but the author provides no directions for those and assumes the reader explicitly knows then already. If this can be done without obscuring the bones in the figure all the better, but it needs fixed.
There is also a bit of added explanation needed around line 210 (see annotated PDF) as to what kind of expansion the author is intending to interpret.
These should all be fairly easy and fairly small fixes though, and the new record will be a nice addition to the literature.

As mentioned above, there are some issues, particularly with the use of articles such as 'the' and 'a' that need fixed. I have mentioned a number of these in the PDF, but I likely did not catch everything, and did not attempt to use my time to catch everything anyway.
Author Response
Rev#3 comments and answers
( ) I would not like to sign my review report
(x) I would like to sign my review report
Quality of English Language
(x) The English could be improved to more clearly express the research.
( ) The English is fine and does not require any improvement.
|
Yes |
Can be improved |
Must be improved |
Not applicable |
|
|
Does the introduction provide sufficient background and include all relevant references? |
(x) |
( ) |
( ) |
( ) |
|
Is the research design appropriate? |
(x) |
( ) |
( ) |
( ) |
|
Are the methods adequately described? |
( ) |
(x) |
( ) |
( ) |
|
Are the results clearly presented? |
( ) |
(x) |
( ) |
( ) |
|
Are the conclusions supported by the results? |
(x) |
( ) |
( ) |
( ) |
|
Are all figures and tables clear and well-presented? |
( ) |
(x) |
( ) |
( ) |
Comments and Suggestions for Authors
The author does a good job at their intended goal. This represents a new record expanding the known range of a species, and providing at least some context for it.
Having said that, there are a few small ways the paper could be improved, and a few things that need correcting. I have included an attached annotated PDF pointing these things out more directly in the text.
Overall, there are a few places that need fixed grammatically. These mostly deal with the use of articles, and, more often than not, are from missing articles. I have pointed some of these out in the PDF.
I agree with the identification, so the main thing is providing some clarification for the reader in the paper.
In particular, there is an instance that needs clarification in the Materials and Methods section on what is meant by 'no pictures taken.'
Additionally, and perhaps what needs the most work, are a few things in the Results section. These are in the annotated PDF, but there are several things that either need to be further explained or clarified. This includes Figure 3, which, while I know some features are pointed out in Figure S1, still needs to have some edits done. Some of these features should be pointed out, or labeled, or both, and this is made more clear that the caption states the parts of the figure are supposed to highlight particular features, but the author provides no directions for those and assumes the reader explicitly knows then already. If this can be done without obscuring the bones in the figure all the better, but it needs fixed.
There is also a bit of added explanation needed around line 210 (see annotated PDF) as to what kind of expansion the author is intending to interpret.
These should all be fairly easy and fairly small fixes though, and the new record will be a nice addition to the literature.
Comments on the Quality of English Language
As mentioned above, there are some issues, particularly with the use of articles such as 'the' and 'a' that need fixed. I have mentioned a number of these in the PDF, but I likely did not catch everything, and did not attempt to use my time to catch everything anyway.
Answer:
I thank the reviewer for constructive feedback and positive evaluation of the manuscript. Comments from the annotated PDF were implemented. I revised the manuscript for grammatical accuracy, with particular attention to article use (“the,” “a,” “an”). These corrections have been implemented throughout the revised version.
To clarify the Materials and Methods section on what is meant by “no pictures taken.”, I added text to Discussion, Also required by other Reviewer:
As for the further steps, additional trapping in the site is required to confirm presence of S. isodon and taking pictures of the animal to assess pelt color and body proportions. The genetic identification of newly trapped specimens is recommended, as well as re-identification of the shrew skull collection at State Scientific Research Institute Nature Research Centre, Lithuania. The skull collection of small mammals is currently maintained by the Mammal Ecology Laboratory. In the future, the Nature Research Centre plans to complete the installation of the Collections Department for the storage of all available collections.
We stand by the publisher's requirement that the photos should not be modified, so the characteristics typical of the species are not additionally indicated in Figure 3, referring the reader to Fig. S1, where the necessary areas are clearly marked without obscuring the characteristics in Fig. 3. Labels and scale were added to Figure 3.
As now written near Line 210, the intended meaning was not ecological expansion, but rather geographical expansion.
I would like to express my gratitude once more to the reviewer for their constructive feedback, which has led to substantial enhancements in the clarity and presentation of my manuscript.